# SELF-SUPERVISED PRIVACY PRESERVATION VIA LATENT ANONYMIZATION FOR GENERALIZABLE VIDEO UNDERSTANDING

## ABSTRACT

The rapid advancements in large video models have unlocked new horizons in video understanding, enhancing applications in various domains such as surveillance, healthcare, and entertainment. However, these models often compromise individual privacy by inadvertently revealing sensitive private information such as skin color and gender. Existing privacy preservation methods are often limited in their scope and tailored to specific downstream tasks. Since current methods directly apply an anonymization function to the input pixel space, they demand extensive computational resources due to the retraining of the utility video model. To address these challenges, we propose a novel approach that shifts privacy-preserving anonymization from the input pixel space to the latent feature space, significantly reducing computational costs and enabling deployment in large foundational video models. Our method employs a self-supervised privacy budget in the latent space by minimizing the mutual information between static clip features. This approach notably allows, for the first time, supervision from downstream tasks such as anomaly detection and temporal action detection through collaborative co-training. Furthermore, we introduce a latent consistency loss to maintain the utility video model's multitask generalization capabilities and prevent single task overfitting. Our extensive evaluations demonstrate a significant ($\approx$**29%**) reduction in privacy leakage while maintaining near peak (within **1%**) utility performance across various downstream tasks: Action Recognition (Kinetics400, UCF101, HMDB51), Temporal Action Detection (THUMOS14), and Anomaly Detection (UCF-Crime). Moreover, we propose new protocols for assessing gender bias in action recognition models, demonstrating that our method effectively mitigates such biases and promotes equitable video understanding.

## 1 INTRODUCTION

Video understanding encompasses a wide range of problem formulations, including action recognition, anomaly detection, and temporal action localization. It holds significant potential for real-world applications such as patient behavior monitoring, sports analytics, robotics, industrial automation, surgical videos, and surveillance. Recent advancements in video understanding have aimed at developing single models capable of addressing multiple video tasks, leading to the creation of video foundational models such as InternVideo Wang et al. (2022), VideoMAEv2 Wang et al. (2023), V-JEPA Bardes et al. (2023), NMS Dave et al. (2024) etc. These models offer a deeper understanding of the intrinsic nature of videos, thereby enhancing their applicability in real-world scenarios.

However, the utility of video understanding models is accompanied by the risk of privacy breaches. Features extracted from pre-trained video encoders can reveal sensitive information about individuals in the videos, such as skin color, gender, and clothing Fioresi et al. (2023). This information can be exploited maliciously, underscoring the urgent need to prevent such privacy leaks from powerful contemporary video models.

Several prior privacy-preserving methods have been proposed to address this issue, including approaches like SPAct Dave et al. (2022b), STPrivacy Li et al. (2023b), TeD-SPAD Fioresi et al. (2023) and VITA Wu et al. (2020). These methods excel at learning anonymization functions that maintain

the utility performance of targeted video downstream task while reducing privacy leakage. However, they encounter several challenges. For instance, after learning the anonymization function, they require full fine-tuning of the utility video encoder model, which demands substantial computational resources and is impractical for scaling to large-scale models such as ViT-H, ViT-L of V-JEPA, and InternVideo. Additionally, these methods often focus on specific downstream video tasks, such as action recognition in SPAct and anomaly detection in TeD-SPAD, limiting their generalizability.

To address the challenges of scalability and generalization in video model anonymization, we introduce a novel approach termed **SPLAVU**, which stands for **S**elf-supervised **P**rivacy-preservation via **L**atent **A**nonymization for general **V**ideo **U**nderstanding. SPLAVU aims to prevent privacy leakage while maintaining downstream performance. Our method operates in the latent space of a utility model, enabling the training of a proposed Anonymizing Adapter Module (AAM) on a frozen utility video model. By eliminating the need for full fine-tuning, SPLAVU is feasible for large video models and adaptable to various downstream tasks.

We adopt the minimax optimization strategy, a proven method in prior privacy works Dave et al. (2022b); Wu et al. (2020); Fioresi et al. (2023), to the latent space. Our joint optimization strategy balances privacy and proxy-utility branches by minimizing the mutual information between two static clips derived from the same video, following the self-supervised privacy objective Dave et al. (2022b). The latent formulation enables a novel co-training paradigm, where the anonymizer is trained to maintain the performance across *multiple* downstream video understanding tasks, not just for a single task. Additionally, to preserve the generalization capability of the utility model on unseen tasks, we introduce a *latent consistency loss*.

Our anonymization method integrates seamlessly with multiple state-of-the-art methods for various downstream tasks. We demonstrate that SPLAVU outperforms prior privacy preservation methods by a significant margin in numerous downstream video understanding tasks. Additionally, SPLAVU exhibits data efficiency; even when trained on a small-scale dataset like HMDB51 Kuehne et al. (2011), it generalizes to multiple tasks without compromising downstream performance and privacy tradeoff. Beyond privacy protection, we address the emerging issue of human-attribute related bias in video understanding, which has been largely unexplored. For instance, a model might predict specific actions based on gender stereotypes, such as associating a perceived female subject with hands near her face as applying makeup or brushing hair, even with nothing in hand. For the first time, we propose protocols to evaluate gender bias in action recognition models and demonstrate our method's effectiveness in mitigating such bias.

Our contributions can be summarized as follows:

- We introduce a novel latent privacy preservation method that is effective across various downstream video understanding tasks using a collaborative multitask co-training protocol, unlike prior work that focused solely on one specific task.

- To enable anonymization in the latent space, we propose a static-clip-based self-supervised privacy budget objective. We also introduce a latent consistency loss to preserve the generalization capability of the utility model.

- We conduct extensive evaluations of our method on multiple downstream tasks. Our method achieves notable decrease in privacy leakage by approximately **29%**, while preserving performance across a range of downstream tasks, including Action Recognition (UCF101, HMDB51, Kinetics400), Temporal Action Detection (THUMOS14), and Anomaly Detection (UCF-Crime).

- Our extensive ablation studies demonstrate the data efficiency of our method when trained on a small-scale dataset and its applicability across various video backbones.

- We propose new protocols to assess gender bias in existing action recognition models and demonstrate that our method effectively mitigates this bias.

## 2 RELATED WORKS

Video understanding spans tasks like action recognition, temporal action localization, and weakly-supervised anomaly detection. Various datasets have been introduced Carreira & Zisserman (2017);

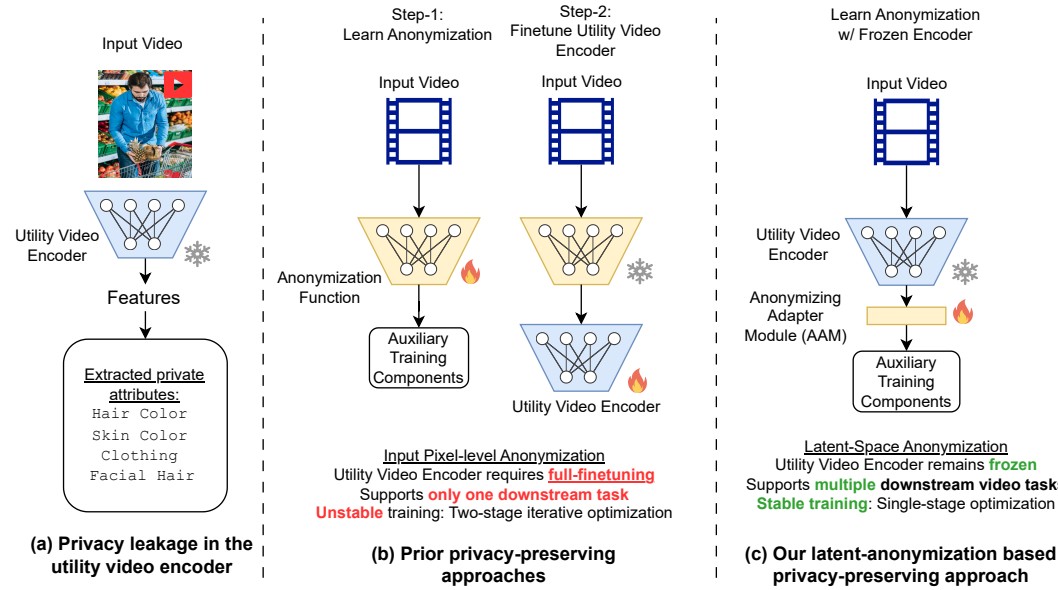

Figure 1: **Motivation (a)** illustrates privacy leakage in pretrained video encoders (*e.g.*, VideoMAE, V-JEPA) by an attacker. **(b)** depicts previous privacy-preserving approaches Fioresi et al. (2023); Dave et al. (2022b); Wu et al. (2020), which involve a two-step training process: first learning an anonymization function to anonymize input videos, then retraining the utility model with the anonymized videos. **(c)** presents our method of joint optimization for the anonymization process in latent space, where we directly learn an adapter module on the utility video encoder, offering numerous benefits as outlined in the figure text.

Diba et al. (2020); Goyal et al. (2017b); Zhao et al. (2019), and recent advancements include self-supervised Jenni & Jin (2021); Dave et al. (2024; 2022a); Thoker et al. (2023) and foundational models Bardes et al. (2024); Wang et al. (2023; 2022) capable of handling multiple video understanding tasks, enhancing versatility and generalizability.

**Privacy Preservation in Video Understanding** Recent efforts in video action recognition have addressed visual privacy concerns. Many studies have aimed to protect visual privacy at the time of data capture by utilizing non-intrusive sensors such as thermal imaging, depth cameras, and event cameras Luo et al. (2018); Hinojosa et al. (2022); Kim et al. (2022); Ahmad et al. (2022; 2023). In this study, we focus exclusively on models using standard RGB cameras. Initial approaches involved reducing the resolution of input data Ryoo et al. (2017); Dai et al. (2015); Liu & Zhang (2020) or employing object detection for targeted obfuscations Ren et al. (2018); Zhang et al. (2021). However, recent research indicates that these methods often fail to balance utility and privacy effectively Wu et al. (2020); Dave et al. (2022b); Fioresi et al. (2023). Wu et al. (2020) showcased an adversarial training framework where a U-Net Ronneberger et al. (2015) modifies input frames to decrease the accuracy of private attribute prediction while preserving action recognition performance. Dave et al. (2022b) proposed a self-supervised variant that focuses on reducing mutual information instead of relying on sensitive privacy labels. Fioresi et al. (2023) adopts the self-supervised privacy objective from Dave et al. (2022b) for the video anomaly detection task.

Compared to the prior input-level anonymization methods our latent-space anonymization method differs in two key aspects: (1) previous methods are tailored to specific downstream tasks, such as action recognition in Dave et al. (2022b) and anomaly detection in Fioresi et al. (2023), while our approach aims to preserve privacy across various downstream video understanding tasks, (2) unlike prior methods, our method does not necessitate the retraining of the video model, thus providing computational efficiency for anonymizing even large-scale video foundation models.

**Bias Mitigation** Computer vision tasks often struggle with spurious correlations Geirhos et al. (2018; 2020), where models rely on irrelevant information to make decisions, such as using background cues for action recognition instead of focusing on subjects' movements Ding et al. (2022);

Zou et al. (2023). Unfortunately, biases across a variety of protected demographic attributes, such as perceived gender, skin color, and age Zhao et al. (2017); Stock & Cisse (2017); Buolamwini & Gebru (2018); Wilson et al. (2019); Prabhu & Birhane (2020); Tong & Kagal (2020); Steed & Caliskan (2021); Gustafson et al. (2023) have been found in vision-based tasks. These biases not only skew model performance but can also perpetuate harmful stereotypes. One common method for bias mitigation is to utilize adversarial training Goodfellow et al. (2014); Xie et al. (2017); Zhang et al. (2018). When targeting a sensitive attribute such as perceived gender, these methods involve the use of a critic model to predict the sensitive attribute as the adversary Beutel et al. (2017); Wang et al. (2019). Barbano et al. (2021) explored the relationship between debiasing and privacy preservation, finding that there exists a subset of privacy preservation methods that are suitable for debiasing, giving promise to privacy preservation as a form of debiasing. In contrast to the image domain, biases in the video domain have not been as extensively studied. While a few papers Choi et al. (2019); Li et al. (2023a) address and mitigate scene bias in action recognition tasks, they overlook biases related to human attributes. Motivated by this gap, we introduce, for the first time, protocols to assess gender bias in action recognition. Our findings demonstrate that our self-supervised privacy preservation method, even without an explicit bias-related objective, effectively generalizes in mitigating gender bias.

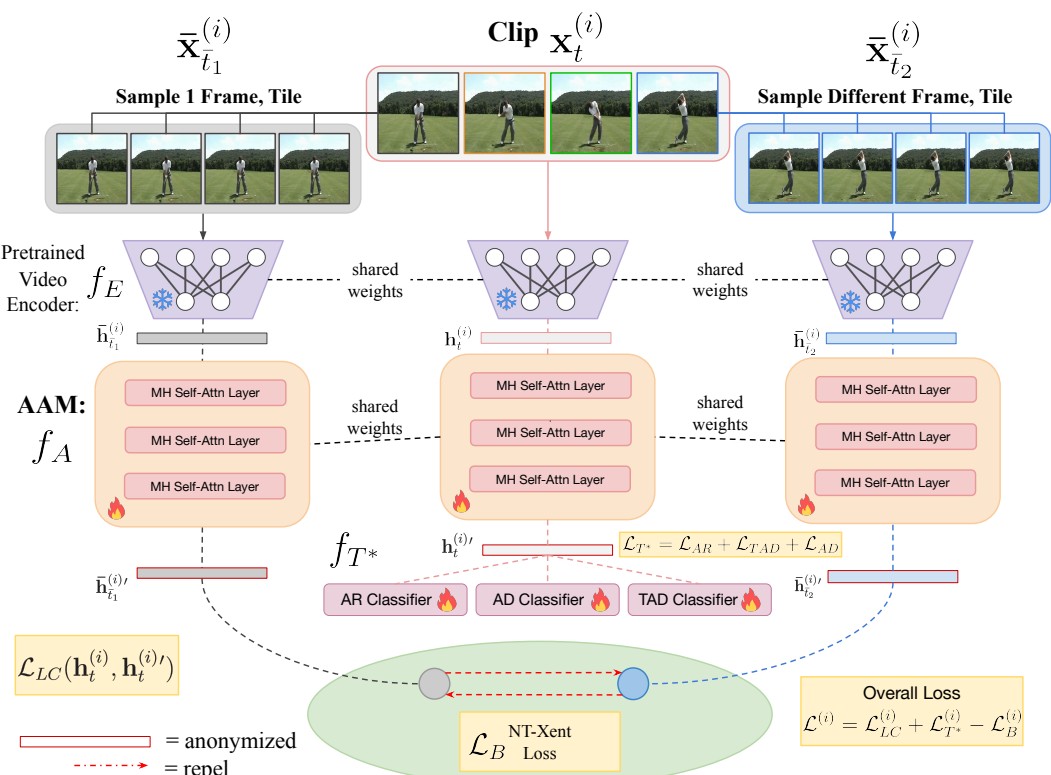

Figure 2: Workflow illustrating the SPLAVU training process. The process begins with a video clip $\mathbf{x}_t^{(i)}$, from which two random frames are sampled to create static clips. All clips are passed through the pretrained, frozen video encoder $f_E$ to extract latent features, which are further processed by our Anonymization Adapter Module (AAM) $f_A$. The feature from the regular clip $\mathbf{x}_t^{(i)}$ is used for the latent consistency loss and passed through the set of task-specific classifier heads $f_{T^*}$, where they are trained using standard utility losses. The two static clips $(\bar{\mathbf{x}}_{\bar{t}1}^{(i)}, \bar{\mathbf{x}}_{\bar{t}2}^{(i)})$ are utilized in a mutual information minimization objective. Gradients are backpropagated through the AAM in a combined backward pass. A complete training algorithm is provided in [Appendix Sec. D](Appendix Sec. D).

## 3 METHOD

### 3.1 PROBLEM FORMULATION

In this work, we consider handling sensitive issues in video understanding tasks from the dual perspective of privacy preservation and bias mitigation.

**Privacy Preservation** We propose a novel privacy-preserving framework that handles multiple utility tasks across diverse video datasets. Our framework is designed to maintain the high performance of a frozen video encoder across tasks while enforcing robust privacy constraints. Specifically, we consider video datasets that span action recognition ($\mathbb{D}_{reco}$), temporal action detection ($\mathbb{D}_{tad}$), and anomaly detection ($\mathbb{D}_{anomaly}$). Each dataset $\mathbb{D}$ contains $N$ video samples, represented as $\{\mathbf{x}^{(i)}, \mathbf{y}^{(i)}\}_{i=1}^{N}$, where $\mathbf{x}^{(i)}$ is a video instance, and $\mathbf{y}^{(i)}$ is its corresponding task-specific label. We define the set of utility tasks as $\{T_{AR}; T_{TAD}; T_{AD}\} \in T^*$, and introduce a budget private attribute prediction task, denoted as $B$. The framework starts with an off-the-shelf video encoder model $f_E$, left completely frozen. The overall goal of $f_A$ is threefold: (1) to maintain the performance of $f_E$ across the set of defined utility tasks $T^*$, (2) to simultaneously reduce the performance on budget private attribute prediction task $B$, and (3) to preserve the general capabilities of $f_E$ such that performance is maintained on *unseen* tasks. This privacy preservation framework is outlined via the following criteria:

*Criterion-1*: Across each utility task, the performance should be retained. Specifically, for task $T^n$, the utility loss $\mathcal{L}_{T^n}$ after anonymization should be approximately equal to the loss before.

$$\sum_{n}^{|T^*|} (\mathcal{L}_{T^n}(f_{T^n}(f_A(f_E(X))), Y), T^n) \approx \sum_{n}^{|T^*|} (\mathcal{L}_{T^n}(f_{T^n}(f_E(X)), Y), T^n). \tag{1}$$

*Criterion-2*: The anonymized encoded features are directly used to compute budget loss $\mathcal{L}_B$ for budget task $B$, which should greatly increase after anonymization.

$$\mathcal{L}_B(f_A(f_E(X))) \gg \mathcal{L}_B(f_E(X)). \tag{2}$$

*Criterion-3*: The anonymization function should not drastically alter the latent features of encoder $f_E$. Hence, we define a latent consistency objective ($\mathcal{L}_{LC}$) as follows.

$$min \, \mathcal{L}_{LC}(f_A(f_E(X)), f_E(X)) \, || \, f_A(f_E(X)) \approx f_E(X), \tag{3}$$

where $||$ denotes OR, as both expressions satisfy the desired condition. A system that fulfills these criteria achieves an effective balance between utility and privacy.

**Perceived Gender Bias** In the standard bias evaluation protocol, we are given a video dataset $\mathbb{D}_{reco} = \{(\mathbf{x}^{(i)}, \mathbf{y}^{(i)})\}_{i=1}^{N_{IID}}$, where $\mathbf{x}^{(i)}$ is the $i$th video instance, $\mathbf{y}^{(i)}$ is the associated action label, and $N_{IID}$ is the number of in-distribution dataset instances. After training, performance is evaluated on an unseen bias test set $\mathbb{D}_{reco-OOD} = \{(\mathbf{x}^{(i)}, \mathbf{y}^{(i)})\}_{i=1}^{N_{OOD}}$, where $N_{OOD}$ is the number of out-of-distribution instances. The aim of any debiasing technique is to learn generalizable features of $\mathbb{D}_{IID}$ such that performance is maximized on $\mathbb{D}_{OOD}$ without compromising IID performance.

When considering gender information, our in-distribution video dataset is now formulated as $\mathbb{D}_{reco} = \{(\mathbf{x}^{(i)}, \mathbf{y}^{(i)}, \mathbf{g}^{(i)})\}_{i=1}^{N_{IID}}$, where $\mathbf{g}^{(i)} \in \{male, female\}$ is the associated socially-perceived binary gender label. We acknowledge that this binary formulation is not ideal and not inclusive of all gender categories. The bias evaluation test set also includes label $\mathbf{g}$ in order to evaluate subclass performance.

### 3.2 ANONYMIZATION FRAMEWORK

This section describes full anonymization framework and training. The framework consists of 3 major components: (1) a frozen video encoder backbone $f_E$, (2) an anonymization function adapter $f_A$, which modifies the latent features while retaining the original shape, and (3) a set of utility classifier heads $\{f_{T_{AR}}; f_{T_{TAD}}; f_{T_{AD}}\} \in f_{T^*}$ for a predefined set of tasks.

**Network Initialization** To start, we initialize $f_A$ to act as an identity function. This involves a brief pretraining phase where $f_A$ learns to reconstruct the latent features from $\mathbb{D}_{reco}$ using an $\ell_1$

loss. The video encoder model $f_E$ is initialized with off-the-shelf weights of Kinetics400 Carreira & Zisserman (2017) pretraining. Each $f_T$ classifier head matches a standard architecture for the provided task. For stability, these are initialized through non-anonymized training on their respective utility tasks. For action recognition, $f_{T_{AR}}$ is a simple linear layer. For temporal action detection and anomaly detection, architectures from TriDet Shi et al. (2023) and MGFN'Chen et al. (2023) respectively are utilized as $f_{T_{TAD}}$ and $f_{T_{AD}}$.

## Anonymization Training

The training process consists of a minimax optimization between a budget privacy loss $\mathcal{L}_B$ and a collection of standard utility losses $\mathcal{L}_{T^*}$, regularized by a proposed latent consistency loss $\mathcal{L}_{LC}$.

**Collaborative Utility Objectives** To retain the action understanding capabilities of the pretrained model, we employ a co-training framework where multiple tasks collaborate to optimize performance. The action classifier head, $f_{T_{AR}}$, is trained using the standard cross-entropy loss:

$$\mathcal{L}_{AR}^{(i)} = -\sum_{c=1}^{N_C} \mathbf{y}_c^{(i)} \log \mathbf{p}_c^{(i)}, \tag{4}$$

where $N_C$ denotes the number of action classes in $\mathbb{D}_{action}$, $\mathbf{y}_c^{(i)}$ is the ground-truth label, and $\mathbf{p}_c^{(i)}$ is the prediction vector from the utility classifier head $f_{T_{AR}}$.

To ensure consistent performance across other utility tasks, we integrate the training objectives of the state-of-the-art approaches for temporal action detection and anomaly detection. Specifically, TriDet Shi et al. (2023) is utilized for $f_{T_{TAD}}$ and $\mathcal{L}_{TAD}$, while MGFN Chen et al. (2023) is employed for $f_{T_{AD}}$ and $\mathcal{L}_{AD}$. More detailed information can be found in Appendix Sec. B.

The utility losses from these tasks are combined and jointly optimized through the following :

$$\mathcal{L}_{T^*}^{(i)} = \omega_{AR}\mathcal{L}_{AR} + \omega_{TAD}\mathcal{L}_{TAD} + \omega_{AD}\mathcal{L}_{AD}, \tag{5}$$

where $\omega$ represents a hyperparameter controlling the relative weight of each task's loss objective. In most experiments, we set $\omega_{AR} = \omega_{TAD} = \omega_{AD} = 1$ to balance the contributions across tasks.

**Budget Privacy Objective** As opposed to previous works Wu et al. (2020); Dave et al. (2022b); Fioresi et al. (2023), we do not use the disjoint image encoder model to enforce the privacy objective. Instead, the video encoder model itself is used to process clip frames (tiled to match a standard clip shape, see Figure 2). These static clip features are then directly utilized in the budget SimCLR contrastive NT-Xent Chen et al. (2020) loss $\mathcal{L}_B$, defined as follows:

$$\mathcal{L}_B^{(i)} = -log \frac{d(\bar{\mathbf{h}}_{t_1}^{(i)}, \bar{\mathbf{h}}_{t_2}^{(i)})}{\sum_{j=1}^{N}[\mathbb{K}_{[j\neq i]}d(\bar{\mathbf{h}}_{t_1}^{(i)}, \bar{\mathbf{h}}_{t_1}^{(j)}) + d(\bar{\mathbf{h}}_{t_1}^{(i)}, \bar{\mathbf{h}}_{t_2}^{(j)})]}, \tag{6}$$

where $\bar{\mathbf{h}}_t^{(i)}$ represents a static clip sampled from video $\mathbf{x}^{(i)}$ at time $t$, $d(u,v) = exp(u^T v/(\|u\|\|v\|\tau))$ computes the similarity between the input vectors with temperature parameter $\tau$. $\mathbb{K}_{[j\neq i]}$ is an indicator function that equals 1 when $j \neq i$. This loss is used to maximize the similarity between the input static clips, but we reverse the gradient, resulting in the objective destroying mutual information between these static clips instead.

**Latent Consistency Objective** The primary motivation behind introducing the latent consistency loss is to ensure that the anonymization learned by the model remains generalizable and is not biased toward the specific utility task(s) it is trained on. Without this constraint, the anonymization process inadvertently overfits to the proxy-utility tasks (see Table 5), compromising its effectiveness on unseen tasks. To mitigate this, we employ a reconstruction-based latent consistency loss that encourages the model to preserve important latent features while still achieving privacy preservation:

$$\mathcal{L}_{LC}^{(i)} = \|f_E(\mathbf{x}^{(i)}) - f_A(f_E(\mathbf{x}^{(i)}))\|_2^2, \tag{7}$$

where $\mathbf{x}^{(i)}$ is the input video clip and $\| \cdot \|_2^2$ is the $\ell_2$ distance. This objectives ensures that the anonymization does not shift $f_E$ features completely into a new space overfit to the training tasks.

**Overall Minimax Training Objective** Due to our use of just a single encoder model, we can avoid the two-step training process from previous works without experiencing collapse. Instead, the $f_A$ and $f_T$ models are jointly optimized utilizing the compound loss as follows:

$$\mathcal{L}^{(i)} = \omega_{LC} * \mathcal{L}_{LC}^{(i)} + \omega_T * \mathcal{L}_{T^*}^{(i)} - \omega_B * \mathcal{L}_B^{(i)}, \tag{8}$$

where $\omega_R$, $\omega_T$, and $\omega_B$ are weights to control the strength of each objective. After this training, we are left with a lightweight anonymization adapter $f_A$ that can be appended to the off-the-shelf video encoder model $f_E$ for use in a variety of downstream tasks.

**Anonymizing Adapter Module (AAM)** To carry out the anonymization function $f_A$, we propose to move away from the common input-level modification techniques, instead applying our proposed latent anonymizing adapter module (AAM). Given latent feature $\mathbf{h}^{(i)} = f_E(\mathbf{x}^{(i)})$, the AAM module is trained to modify $\mathbf{h}^{(i)}$ with the above loss objective . We utilize a multi-head self-attention based transformer encoder for our adapter. A design choice ablation can be found in Appendix Sec. C.

## 4 EVALUATION PROTOCOLS

To ensure that our anonymization method preserves the utility of the original off-the-shelf encoder across multiple tasks, we evaluate its performance comprehensively. Previous work Fioresi et al. (2023) has shown that existing anonymization methods, which typically use action recognition as the proxy utility task, significantly degrade performance on alternate downstream tasks. However, the original pretrained models are known to demonstrate strong performance in areas like temporal action detection (TAD) and anomaly detection (AD). Therefore, we assess the learned features across *five distinct tasks* to thoroughly evaluate their effectiveness post-anonymization.

### 4.1 PRIVACY EVALUATION

First, we utilize an established privacy preservation protocol to ensure that $f_A$ removes sensitive-attribute related information. Even though we work with with action-focused video understanding models, information related to sensitive attributes is still carried through the backbone encoder. This is empirically shown through performance on a private attribute prediction task using the VISPR dataset Orekondy et al. (2017). While this is an image-based protocol, we are still able to use it by repeating each image to form static clips. A classifier is trained on these representations using the same protocol as previous video privacy works Wu et al. (2020); Dave et al. (2022b); Fioresi et al. (2023). The attribute labels are multi-label, so performance is measured by mean average precision across classes (cMAP). Since our goal is to enhance privacy—not to accurately predict attributes—a lower performance on this task indicates better privacy preservation.

### 4.2 UTILITY VIDEO TASK EVALUATION

**Action Recognition** Action recognition involves analyzing spatio-temporal features of a video to classify the actions based on a predefined set of categories. Our anonymization framework uses action recognition as its proxy utility task, carried out on multiple different datasets $\mathbb{D}_{reco}$, namely Kinetics400 Carreira & Zisserman (2017), UCF101 Soomro et al. (2012), and HMDB51 Kuehne et al. (2011). Evaluation is top-1 accuracy on 5 evenly spaced clips from each testing video.

**Temporal Action Detection** Temporal action detection (TAD) is a task that involves identifying the specific time intervals within an untrimmed video where particular actions occur. TAD utilizes features from a Kinetics-pretrained video encoder model. Given $\mathbb{D}_{tad}$, $f_A$ is used to generate anonymized feature set $\mathbb{F}_{tad} = \{ f_A(f_E(X^{(i)})) \mid \forall X^{(i)} \in \mathbb{D}_{tad} \}$. Our TAD evaluation uses THUMOS14 Jiang et al. (2014) as $\mathbb{D}_{tad}$. We choose one of the recent state-of-the-art methods, TriDet Shi et al. (2023) with default hyperparameters to evaluate using mean Average Precision (mAP).

**Weakly-Supervised Anomaly Detection** Weakly supervised anomaly detection (WSAD) involves localizing the timestamps of anomalous (unexpected) events given long, untrimmed videos and only video-level labels. Our evaluation for WSAD uses UCF-Crime Sultani et al. (2018) as $\mathbb{D}_{anomaly}$. Given $\mathbb{D}_{anomaly}$, $f_A$ is used to create anonymized feature set $\mathbb{F}_{anomaly} = \{ f_A(f_E(X^{(i)})) \mid \forall X^{(i)} \in \mathbb{D}_{anomaly} \}$. A recent state-of-the-art method MGFN Chen et al. (2023) is used with default hyperparameters. Final evaluation is given as frame-level ROC AUC percentage.

### 4.3 GENDER PRESENTATION BIAS PROTOCOLS

**Proposed NTU Bias Evaluation** We further verify the anonymization performance by evaluating on our proposed attribute video bias protocols. The NTU 60 Shahroudy et al. (2016) action recognition dataset is carefully curated to have minimal scene and subject biases as each actor performs each action multiple times in different scenes. Given that each video is carefully labeled with subject ID, we can introduce an artificial bias by controlling the subclass ratios across actions. A gender ratio of 95% is set for all but one action, where the typical gender ratio is inverted to create a spurious shortcut for the model. This is done for each gender, resulting in two subsets: NTU-Bias-F and NTU-Bias-M. More detailed information about protocol creation can be found in Appendix Sec. A.

**Toyota Smarthome Bias Evaluation** Unlike NTU 60, the Toyota Smarthome (TSH) ADL dataset is less balanced and represents a real-world scenario with elderly individuals performing unscripted daily activities. Each video is labelled with a subject ID, allowing for robust evaluation of perceived gender biases without the need for a pretrained gender classifier. Here, we look at the performance of each gender subclass. A model is considered less biased if the baseline gap between the performance of each gendered subclass is reduced.

## 5 EXPERIMENTS

Dataset details and implementation details can be found in Appendix Sec. A and B, respectively.

Table 1: Quantitative results of various anonymization methods on various downstream tasks. **Bold** indicates best results.

| Anonymization Method | Network | Privacy | Action Reco. | Temp. Act. Detection | Anomaly Detection | Overall Score |
|---|---|---|---|---|---|---|
| | | VISPR | Kin400 | THUM14 | UCFCrime | |
| | | cMAP (↓) | Top-1 (↑) | mAP(↑) | AUC (↑) | (↑) |
| Raw Videos | I3D | 63.64 | 50.98 | 49.88 | 77.68 | 71.91 |
| SPAct | | 52.71 | 46.93 | 29.98 | 73.93 | 73.18 |
| TeD-SPAD | | 42.21 | 47.20 | 32.08 | 74.81 | 81.87 |
| **Ours** | | 39.79↓37.5% | 50.84↓0.3% | 47.96↓3.8% | 75.82↓2.4% | **88.81** |
| Raw Videos | VidMAE | 70.25 | 74.83 | 60.19 | 84.72 | 77.25 |
| **Ours** | | 49.92↓28.9% | 74.23↓0.8% | 60.50↑0.5% | 84.33↓0.5% | **92.51** |
| Raw Videos | VJEPA | 72.44 | 77.03 | 66.06 | 84.12 | 77.47 |
| **Ours** | | 51.42↓29.2% | 76.62↓0.5% | 66.30↑0.4% | 84.81↓1.1% | **93.37** |
| Raw Videos | VidMAEv2 | 75.69 | 91.24 | 67.24 | 85.73 | 79.29 |
| **Ours** | | 50.37↓33.5% | 91.01↓0.3% | 65.59↓2.4% | 83.70↓2.5% | **97.30** |

### 5.1 MAIN EVALUATION: PRIVACY VS DOWNSTREAM TASK TRADEOFFS

Our evaluation of the proposed method, as outlined in Sec. 4, covers privacy protocols and a variety of downstream tasks. We observe that our approach consistently generalizes well across all tasks, closely maintaining the performance of the raw, unanonymized videos. In contrast, previous methods struggle to preserve performance uniformly across tasks, evident in the temporal action detection results of Dave et al. (2022b); Fioresi et al. (2023). Additionally, when considering an overall score that combines both privacy and average downstream performance (Details about score metric in Appendix Sec. B), our method surpasses the prior best Fioresi et al. (2023) by **6.94%**. Experiments with large VFMs confirm the efficacy and scalability of SPLAVU as we see similar trends in performance for each model, even with increasing feature dimensionality.

## 5.2 Gender Bias Evaluation

The first row of Table 2 shows the performance difference between each gender presentation subclass in the NTU-Bias-F protocol, where the action *brush_hair* is chosen as the gendered shortcut action label. In the baseline, the performance disparity between perceived gender subclasses is an unacceptable large 9.42%. We see that using our proposed latent anonymization method SPLAVU impressively reduces this gap by **42.3%**. The second row includes results for the complimentary protocol NTU-Bias-M, also with the *brush_hair* shortcut class. Interestingly, the baseline subclass performance disparity is less than that of NTU-Bias-F (5.00%), but our method is still capable of reducing this unfair split and improving overall performance.

To confirm that these observations hold true in a real-world setting, we look at the final row of Table 2 to see the performance on the Toyota Smarthome Das et al. (2019) protocol. Notably, our method improves the classifier quality while simultaneously improving the fairness of its decision making. In this realistic scenario with a naturally occurring bias, SPLAVU is able to reduce the gap between perceived gender subclasses by an astonishing **34.0%**.

Table 2: Performance on NTU-Bias-F, NTU-Bias-M, and Toyota SH, split across gender subgroup.

| Dataset | Method | P. Female Acc. (%) | P. Male Acc. (%) | Overall Acc. (%) | Δ Subclass Acc. Reduction (%) |
|---|---|---|---|---|---|
| NTU-Bias-F | Baseline | 46.78 | **56.20** | 51.49 | 0.00 |
| | Ours | **49.91** | 55.35 | **52.63** | **42.3** |
| NTU-Bias-M | Baseline | **55.23** | 50.23 | 52.78 | 0.00 |
| | Ours | 55.07 | **51.04** | **53.06** | **19.4** |
| TSH | Baseline | 63.54 | 68.18 | 65.05 | 0.0 |
| | Ours | **66.13** | **69.65** | **67.27** | **34.0** |

## 5.3 Ablations and Analysis

We utilize the VideoMAE-B model for all ablations. Further details are found in Appendix Sec. C.

**Effect of task-specific anonymization training:** Our important ablation in Table 3 demonstrates the effects of training our anonymizer without specific task heads. Notably, the highlighted cells show impressive generalization to unseen tasks in each experiment with only a minor drop in performance compared to training on the task. For example, looking at row (c) shows $f_A$ training with only temporal action detection as the utility task. Even so, the performance on action recognition and anomaly detection remain within **1.3%** of the non-anonymized score. Across the board, performance is not dependent on having seen the given utility task during training, proving that SPLAVU can effectively *generalize to unseen tasks*.

Table 3: Ablation changing the tasks seen during anonymization training. The checkmark (✓) labels seen tasks, x-mark (✗) and highlighted cells indicate tasks unseen during training.

| | Training Tasks | | | Evaluation Tasks | | | |
|---|---|---|---|---|---|---|---|
| | AR | TAD | AD | VISPR cMAP (↓) | K400 Top-1 Acc. (↑) | THUM14 mAP (%) (↑) | UCF-Crime AUC (%) (↑) |
| (a) | ✗ | ✗ | ✗ | 70.25 | 74.83 | 60.19 | 84.72 |
| (b) | ✓ | ✗ | ✗ | 52.57 | 74.65 | 56.45 | 83.47 |
| (c) | ✗ | ✓ | ✗ | 50.17 | 73.86 | 58.80 | 83.67 |
| (d) | ✗ | ✗ | ✓ | 49.34 | 73.51 | 57.34 | 84.56 |
| (e) | ✓ | ✓ | ✗ | 48.74 | **74.30** | 58.67 | 83.88 |
| (f) | ✓ | ✗ | ✓ | 50.77 | 74.24 | 58.83 | 84.28 |
| (g) | ✗ | ✓ | ✓ | **48.01** | 73.70 | 60.41 | 84.77 |
| (h) | ✓ | ✓ | ✓ | 49.92 | 74.23 | **60.50** | **85.08** |

**Effect of training set scale:** To see if our anonymization method depends on the scale of the training set, we perform all downstream tasks with the varying size of training datasets as shown in Table 4. Here, $f_A$ is trained using action recognition as the only utility task, with the row indicating the anonymization dataset. Each column separately evaluates the frozen anonymizer on the given task/dataset. We see that SPLAVU demonstrates impressive data-efficiency by effectively generalizing to all downstream tasks, even when training from *small-scale* datasets like HMDB51.

Table 4: Evaluating different action recognition training sets for our anonymization process.

| Pretraining Dataset | VISPR cMAP ($\downarrow$) | K400 Top-1 ($\uparrow$) | UCF101 Top-1 ($\uparrow$) | HMDB51 Top-1 ($\uparrow$) | ToyotaSH Top-1 ($\uparrow$) | UCF-Crime AUC ($\uparrow$) | THUM14 mAP ($\uparrow$) |
|---|---|---|---|---|---|---|---|
| Raw | 70.25 | 74.83 | 96.80 | 72.94 | 65.05 | 84.72 | 60.19 |
| K400 | 52.57 | **74.74** | 96.11 | 71.51 | 65.34 | 83.47 | 56.45 |
| UCF101 | **49.64** | 74.49 | **97.01** | 72.68 | 62.29 | 84.14 | 52.18 |
| HMDB51 | 54.35 | 74.55 | 96.56 | **73.92** | 65.82 | **84.52** | **56.50** |
| Toyota SH | 51.58 | 74.35 | 96.09 | 72.42 | **67.27** | 74.92 | 41.30 |

**Different training objectives in anonymization:** Our ablation study examines key training losses of the anonymization process in Table 5. Action recognition is used as the only training utility task to evaluate generalization to unseen tasks. Not unsurprisingly, omitting the utility loss leads to a considerable drop in model performance on all tasks. Excluding the privacy budget objective results in no privacy gains over the baseline, emphasizing its necessity. Furthermore, removing latent consistency loss affects anomaly detection performance but not action recognition, likely because the action classification proxy task preserves action performance. This underscores the importance of latent consistency loss in ensuring the generalization of our anonymization method.



Table 5: Ablation for training objectives.

| $\mathcal{L}_T$ | $\mathcal{L}_B$ | $\mathcal{L}_{LC}$ | VISPR cMAP ($\downarrow$) | HMDB51 Top-1 Acc. ($\uparrow$) | UCF Crime AUC (%) ($\uparrow$) |
|---|---|---|---|---|---|
| ✗ | ✗ | ✗ | 70.25 | 72.94 | 84.02 |
| ✗ | ✓ | ✓ | 45.12 | 4.71 | 60.12 |
| ✓ | ✗ | ✓ | 70.44 | 73.17 | 83.88 |
| ✓ | ✓ | ✗ | 51.70 | 72.88 | 65.62 |
| ✓ | ✓ | ✓ | **54.35** | **73.92** | **84.52** |

Table 6: Ablation for weight of $\mathcal{L}_{LC}$.

| $\omega_{LC}$ | VISPR cMAP ($\downarrow$) | HMDB51 Top1 Acc. ($\uparrow$) | UCF Crime AUC (%) ($\uparrow$) |
|---|---|---|---|
| 0 | 51.7 | 72.88 | 65.62 |
| 1 | 48.96 | 73.27 | 72.19 |
| 10 | 52.5 | 73.4 | 72.58 |
| 100 | **54.35** | **73.92** | **84.52** |
| 1000 | 59.2 | 73.33 | 83.57 |



**Relative weightage of latent consistency objective:** To further investigate the importance of latent consistency loss, we consider varying weights w.r.t. the overall training objective in Table 6. Since we want to ensure generalization across unseen tasks, action recognition is the only training utility task in this experiment. We found more solid support that with increasing the weightage of the latent consistency loss, performance maintains on the action-related utility, however, it significantly increases performance on the unseen anomaly detection task.

## 6 CONCLUSION

We propose an innovative self-supervised privacy-preserving method via a novel formulation of latent-space anonymization called SPLAVU. Our method is the first to enable generalized anonymization for unprecedented performance across various downstream video understanding tasks, including action recognition, anomaly detection, and temporal action detection. It employs a self-supervised privacy budget within the latent space, coupled with a latent consistency loss to maintain the powerful generalization capability of the model. Moreover, the latent formulation enables learning the anonymizer with gradients from each task for a boost in performance, which is impractical with input anonymization. SPLAVU also eliminates the need for extensive retraining of the utility video model, enabling the anonymization of large foundational models with computational efficiency. Our extensive evaluations and ablation studies validate the effectiveness and data efficiency of our approach. Furthermore, our innovative protocols for assessing gender bias contribute to the development of more responsible and unbiased video understanding models.

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

## SUPPLEMENTARY OVERVIEW

Section A: Dataset details
Section B: Implementation details
Section C: Additional experiment details
Section D: Training algorithm

## A   DATASET DETAILS

**Kinetics400 Carreira & Zisserman (2017)** is a large-scale video action dataset of YouTube videos which includes 400 human action classes with at least 400 video clips for each action. Each clip lasts around 10 seconds and is labeled with a single action class. The dataset is widely used for pretraining deep learning models for use in many video understanding tasks.

**UCF101 Soomro et al. (2012)** is an action recognition dataset of realistic action videos consisting of 101 action categories. With over 13,000 videos from various actions and scenes, it provides a diverse set of actions and a broad range of variability in terms of actions, viewpoints, appearances, and backgrounds.

**HMDB51 Kuehne et al. (2011)** is a collection of 6,766 video clips distributed across 51 human action categories, each containing a minimum of 101 clips. The dataset includes a wide range of human actions and is designed for the development and evaluation of action recognition methods.

**NTU RGB+D 60 Shahroudy et al. (2016)** is a large-scale multi view human action recognition dataset complete with RGB video, depth maps, and skeleton joints, and IR sequences. This work only uses the RGB frames. Each of the 40 subjects are recorded completing 60 daily activities from 3 different cameras.

**Toyota Smarthome Das et al. (2019)** is a challenging real-world activity classification dataset captured from 7 independent Kinect v1 cameras. The clips recorded 18 senior subjects performing 31 daily activities in a natural manner. This work only uses the provided RGB frames. The dataset contains high class imbalance, intra-class variation, and duration variance.

**THUMOS14 Jiang et al. (2014)** focuses on temporal action localization in untrimmed videos. It extends the UCF101 dataset with temporal annotations for a subset of the action classes, providing detailed temporal annotations for 20 action classes across 200 validation videos and 213 test videos.

**UCF-Crime Sultani et al. (2018)** is a large-scale dataset of surveillance videos designed for anomaly detection. It consists of 1,900 long and untrimmed videos for a total of 128 hours. The videos contain examples of 13 different real-world anomalies, including burglary, robbery, and fighting, among others, making it suitable for training and evaluating video anomaly detection models.

**VISPR Orekondy et al. (2017)** consists of around 22,000 Flickr images annotated with 68 privacy-related attributes such as gender, age group, skin color, and more. It offers a multi-class classification protocol for assessing private attribute prediction. Table 7 shows the VISPR attribute split used, which we have adopted from Wu et al. (2020); Dave et al. (2022b); Fioresi et al. (2023).

**Proposed NTU Bias Evaluation Details** More formal details for the creation of the proposed perceived gender NTU bias protocol are described here. While the original dataset is balanced in terms of scene and actor, the distribution of actor/video counts are not balanced with respect to perceived gender. To properly evaluate bias mitigation, it is essential to ensure that there are no performance differences stemming from the larger number of male subjects and training videos. The subject IDs are used to first restructure the dataset in an effort to maximize fairness across the gender sub-groups. As such, within themselves, the train and test sets should contain both an even number of male and female subjects AND an even number of videos per action. Formally, lets take the set of subjects $S = \{s_i\}_{i=1}^{N_S}$, where $N_S$ is the number of subjects in the dataset. For each subject $s_i \in S$, there is an associated gender label $\mathbf{g}(s_i)$ where $\mathbf{g}(s_i) \in \{male, female\}$. We set $N_m = N_f = \frac{N_S}{2}$, where $N_m$ and $N_f$ are the number of male and female subjects, respectively. Using the above notation with $\mathbb{D}_{IID}$ abbreviated to $D$, we define $D_m = \{(\mathbf{x}_i, \mathbf{y}_i, \mathbf{g}_i) \in D | \mathbf{g}_i = male\}$

Table 7: Privacy attributes from subset of VISPR Orekondy et al. (2017) labels as used in previous works.

| VISPR1 Wu et al. (2020); Dave et al. (2022b); Fioresi et al. (2023) | |
| --- | --- |
| Label | Description |
| a17_color | skin color |
| a4_gender | gender |
| a9_face_complete | full face visible |
| a10_face_partial | part of face visible |
| a12_semi_nudity | partial nudity |
| a64_rel_personal | shows personal relationship |
| a65_rel_soci | shows social relationship |

and $D_f = \{(\mathbf{x}_i, \mathbf{y}_i, \mathbf{g}_i) \in D | \mathbf{g}_i = female\}$. We set $|D_m| = |D_f| = \frac{|D|}{2}$. With the dataset balanced across subject counts, subject genders, video count per action/gender, and background representation, the model should not have access to simple bias shortcuts.

To directly measure gender presentation bias, we inject an artificial bias related to perceived gender by creating a simple spurious shortcut for the model to follow. Specifically, we control the subclass ratios across all actions, setting $P(\mathbf{g}(s) = male|\mathbf{y}) = 0.95$ and $P(\mathbf{g}(s) = female|\mathbf{y}) = 0.05$, following the correlation strength in Sagawa et al. (2019). However, for one action chosen at random, we flip this ratio, keeping 95% of perceived female videos ($P(\mathbf{g}(s) = female|\mathbf{y}) = 0.95$) and only 5% of perceived male videos ($P(\mathbf{g}(s) = male|\mathbf{y}) = 0.05$). We refer to this subset as NTU-Bias-F. To ensure that the shortcut taking is gender presentation agnostic, we repeat this protocol by swapping the subclasses, creating NTU-Bias-M. We find that swapping this subclass ratio for one action class reduces overall performance and causes a gap in subclass performance.

# B  IMPLEMENTATION DETAILS

All of our code is implemented in PyTorch Paszke et al. (2019). In this section, we provide implementation details regarding network architecture, input preprocessing, hyperparameters, and training schedules.

## B.1  NETWORK ARCHITECTURE

Each video encoder $f_E$ model is left unchanged from the original implementation. The $f_T$ classifier head is a simple linear layer `Linear(d, N)`, where $d$ is the feature vector dimension of $f_E$ and $N$ is the number of classes in $\mathbb{D}_{reco}$. For the private attribute prediction task, a 2-layer MLP is used: `Linear(d, d)` $\rightarrow$ `Linear(d, 7)` with a ReLU activation after the first layer. For the $f_A$ AAM, we ablate different architecture styles (see Table 8). To break it down, we tried a LoRA-based adapter, standard MLPs of different depths, and self-attention based adapters of different depths. The LoRA adapter features a simple downsample-upsample architecture formulated by: $\mathbf{h}+$ `Linear(d, 256)` $\rightarrow$ `(256, d)`, where $\mathbf{h}$ is the output feature embedding from $f_E$. Each MLP layer is composed of a `Linear(d, d)` followed by a ReLU activation and a BatchNorm1D layer, and dropout with a probability of 0.1 during training. The self-attention layers are standard MulitheadAttention blocks with dim $d$ and 8 heads by default.

## B.2  INPUTS AND AUGMENTATIONS

All inputs consist of 16 frame clips sampled with consecutive frames, resized to spatial resolution of $224 \times 224$. For training, only random resized crop and random horizontal flip with probability 50% are utilized. In validation, the short edge is resized to 256, then a center crop of $224 \times 224$ is taken. Standard ImageNet Krizhevsky et al. (2012) mean and standard deviation based normalization is applied in both settings. The input and augmentation protocol is consistent for every $f_E$.

## B.3 TRAINING DETAILS AND HYPERPARAMETERS

Each AAM variation is trained using an $\ell_1$ loss to reconstruct the input features for 100 epochs with the AdamW Loshchilov & Hutter (2017) optimizer and a learning rate of 2e-5. Kinetics400 Carreira & Zisserman (2017) features are used as the train-test set. Privacy evaluation is carried out using supervised training of the predictor MLP for 100 epochs at a learning rate of 1e-3. A learning rate scheduler is based on the loss plateau where it decreases the learning rate to 1/5th.

For anonymization training, the base learning is 1e-4 for both $f_A$ and $f_{T^*}$, corresponding to a batch size of 512, scaled when necessary according to the linear scaling rule Goyal et al. (2017a). By default, $\omega_{LC} = 100$, $\omega_T = 1$, and $\omega_B = 1$ (Main Eq. 8). The anonymization training is carried out for 100 epochs.

## B.4 OVERALL SCORE METRIC

In order to effectively evaluate and compare each model across multiple tasks, we choose to define a weighted overall performance metric. Specifically, for each evaluated model, we take the raw performance scores of each video understanding task and add this with an inverse of the privacy scores, then divide across tasks. The overall score function is defined as follows:

$$S(T^*) = \frac{(3 * (1 - S(\mathbb{D}_{privacy})) + S(\mathbb{D}_{reco}) + S(\mathbb{D}_{tad}) + S(\mathbb{D}_{anomaly}))}{4}, \tag{9}$$

where $S(T)$ represents the performance on a given task. To prioritize the importance of the privacy task in comparison to the other tasks collectively, it is assigned a weight of 3. This weighting strategy ensures that the significance of achieving the privacy objective is on par with the aggregate significance of all other tasks combined. This scoring metric is used in Main Table 1.

## B.5 ADDITIONAL COLLABORATIVE TASK LOSSES

Here we further define the integrated task losses $\mathcal{L}_{TAD}$ and $\mathcal{L}_{AD}$ referenced in Main Paper Sec. 3.

**Temporal Action Detection Loss $\mathcal{L}_{TAD}$ (TriDet Shi et al. (2023)):**

The overall TriDet loss function combines classification and regression components and is defined as:

$$\mathcal{L}_{TAD} = \frac{1}{N_{pos}} \sum_{l,t} \mathbb{1}_{\{c_t^l > 0\}} \left(\sigma_{IoU} \mathcal{L}_{cls} + L_{reg}\right) + \frac{1}{N_{neg}} \sum_{l,t} \mathbb{1}_{\{c_t^l = 0\}} \mathcal{L}_{cls}, \tag{10}$$

where $N_{pos}$ and $N_{neg}$ are the numbers of positive and negative samples, respectively; $\mathbb{1}_{c_t^l > 0}$ is an indicator function that equals 1 if $c_t^l > 0$ (positive sample) and 0 otherwise; $\sigma_{IoU}$ is the temporal Intersection over Union (IoU) between the predicted segment and the ground truth, serving as a weighting factor; $\mathcal{L}_{cls}$ is the classification loss, implemented as the focal loss Ross & Dollár (2017); and $\mathcal{L}_{reg}$ is the regression loss, implemented as the IoU loss Rezatofighi et al. (2019). The weighting factor $\sigma_{IoU}$ emphasizes predictions with higher temporal IoU, ensuring that higher-quality predictions contribute more significantly during training. Positive samples are determined using center sampling, where instants near the center of an action instance are labeled as positive, and others are considered negative.

**Anomaly Detection Loss $\mathcal{L}_{AD}$ (MGFN Chen et al. (2023)):**

The full MGFN loss function is defined as:

$$\mathcal{L}_{AD} = \mathcal{L}_{sce} + \lambda_1 \mathcal{L}_{ts} + \lambda_2 \mathcal{L}_{sp} + \lambda_3 \mathcal{L}_{mc}, \tag{11}$$

where $\lambda_1 = \lambda_2 = 1$ and $\lambda_3 = 0.001$. The base loss $\mathcal{L}_{sce}$ is the standard sigmoid cross-entropy loss:

$$\mathcal{L}_{sce} = -y \log(s^{i,j}) - (1 - y) \log(1 - s^{i,j}), \tag{12}$$

with $y$ as the video-level label ($y = 1$ for anomaly, $y = 0$ for normal) and $s^{i,j}$ as the computed anomaly score for frame $i$ in segment $j$. Following Sultani et al. (2018), it incorporate a temporal smoothness term $\mathcal{L}_{ts}$ and a sparsity term $\mathcal{L}_{sp}$:

$$\mathcal{L}_{ts} = \sum_{i=1}^{n-1} \left( f(V_a^i) - f(V_a^{i+1}) \right)^2, \tag{13}$$

$$\mathcal{L}sp = \sum_{i=1}^{n} f(V_a^i), \tag{14}$$

where $f(V_a^i)$ represents the extracted features for segment $i$ of an anomalous video $V_a$. These terms encourage smooth transitions between sequential segments and promote sparsity in detections.

MGFN introduces a feature amplification mechanism and a magnitude contrastive loss $\mathcal{L}_{mc}$ to enhance feature separability within and between videos, formulated as:

$$\mathcal{L}_{mc} = \sum_{p,q=0}^{B/2} (1-l)(D(M_n^p, M_n^q)) + \sum_{u,v=B/2}^{B} (1-l)(D$$

$$(M_a^u, M_a^v)) + \sum_{p=0}^{B/2} \sum_{u=B/2}^{B} l(Margin - D(M_n^p, M_a^u)), \tag{15}$$

where $B$ is the batch size, $M$ denotes the feature magnitude of the corresponding segment, $l$ is an indicator function, and $D(\cdot, \cdot)$ is a distance function. Refer to Chen et al. (2023) for more details.

## C  ADDITIONAL EXPERIMENTS

**Different Architectures for Anonymizing Adapter Module (AAM):** Our ablation study evaluates different AAM architectures in Table 8, with the baseline showing standard performance without anonymization. The multi-layer perception (MLP) adapter demonstrates moderate privacy enhancement, particularly with increased capacity, while nearly maintaining utility performance. However, the self-attention-based module is superior across the board, finely balancing privacy and utility, making it our Anonymizing Adapter Module of choice. The difference between the encoder having 3 and 5 layers is negligible, as performance appears to plateau with the larger capacity. As such, for more efficient compute without sacrificing performance, we adopt the 3 encoder layer self-attention AAM for the majority of experiments. Self-attention's efficacy is likely due to its ability to prioritize crucial features for anonymization, refining the privacy preservation process.

Table 8: Ablation with different architecture of Anonymizer Adapter Modules.

| Anonymizer Architecture | VISPR cMAP ($\downarrow$) | K400 Top1 Acc. ($\uparrow$) | UCF Crime AUC (%)($\uparrow$) | THUM14 mAP ($\uparrow$) |
|---|---|---|---|---|
| None | 70.25 | 74.83 | 84.72 | 60.19 |
| MLP (1 hidden layer) | 67.51 | 73.39 | 82.13 | 59.42 |
| MLP (3 hidden layer) | 62.92 | 64.84 | 79.63 | 54.60 |
| MLP (5 hidden layer) | 61.92 | 70.03 | 83.47 | 57.34 |
| Self Attention (1 layer encoder) | 50.59 | 72.57 | 82.12 | 58.17 |
| Self Attention (3 layer encoder) | 49.92 | **74.23** | **84.33** | **60.50** |
| Self Attention (5 layer encoder) | **48.56** | 74.08 | 83.54 | 57.46 |

### C.1  ABLATION WITH ATTENTION HEAD COUNTS IN AAM:

In addition to the ablations in the main paper, we show here in Table 9 the effect of changing the number of heads in the MHSA layer of our transformer based AAM. The performance for each variation was very similar, with the middle 8 heads beating out the other variations, providing a solid tradeoff for compute and performance. Our default experiment setup utilizes 8 MHSA heads.

Table 9: Ablation with different number of MHSA Heads.

| Num MHSA Heads | VISPR cMAP ($\downarrow$) | HMDB51 Top1 Acc. ($\uparrow$) | UCF Crime AUC (%) ($\uparrow$) |
|---|---|---|---|
| 4 | 54.78 | 73.79 | 83.73 |
| 8 | **54.35** | **73.92** | **84.52** |
| 16 | 56.74 | 73.73 | 83.99 |

## C.2 COMPARISON WITH OTHER ANONYMIZATION METHODS

Previous work Wu et al. (2020); Dave et al. (2022b); Fioresi et al. (2023) has already shown that the learnable anonymization techniques outperform methods such as downsampling, blurring, and blackening. Table 10 shows a comparison to these techniques. In Downsample-2x and Downsample-4x, the input frames have their resolution reduced by a factor of 2 ($112 \times 112$) and 4 ($56 \times 56$). In Blackening and Blurring, subjects are detected using an object detector to detect human subjects and obfuscated using the same methods as described in Wu et al. (2020); Dave et al. (2022b); Fioresi et al. (2023). We see that none of these techniques achieve an acceptable level of anonymization, and almost all reduce utility more than our SPLAVU method, further demonstrating the capability of our framework.

Table 10: Additional experiments with existing obfuscation techniques. All experiments use I3D as the network backbone.

| Anonymization Method | VISPR cMAP ($\downarrow$) | UCF101 Top1 Acc. ($\uparrow$) | UCF Crime AUC (%) ($\uparrow$) |
|---|---|---|---|
| Raw Videos | 63.64 | 89.25 | 77.68 |
| Downsample-2x | 55.64 | 81.78 | 76.09 |
| Downsample-4x | 52.84 | 66.21 | 68.12 |
| Blurring | 58.68 | 83.90 | 75.69 |
| Blackening | 56.36 | 68.62 | 73.91 |
| Ours | 40.42 | 90.48 | 75.82 |

Table 11 shows the results of our proposed method on PA-HMDB Wu et al. (2020) compared to the baseline model on raw data.

Table 11: PA-HMDB51 results, using VideoMAE as $f_E$.

| Method | Privacy cMAP ($\downarrow$) | Action Top-1 Acc ($\uparrow$) |
|---|---|---|
| Baseline | 69.9 | 80.19 |
| Ours | **62.6** | **84.47** |

## C.3 ADDITIONAL EXPERIMENTS WITH LARGE FOUNDATION MODELS

Due to the low compute cost and focus on maintaining the capabilities of powerful models, our SPLAVU framework is able to scale up to the largest video foundational models currently available. Table 12 demonstrates the high privacy-utility tradeoff achieved by our method using Video-MAEv2 Wang et al. (2023) and InternVideo Wang et al. (2022). In these experiments, action recognition performance was exactly maintained, and private attribute prediction was dropped more than for the smaller models, with only a modest reduction in temporal action detection performance.

### C.3.1 TRAINING COMPUTE

One of the many benefits of our SPLAVU framework is its very low compute/training cost. Table 13 shows the overall count of trainable parameters for previous frameworks compared to our AAM. For VideoMAE-Base, our the SPLAVU framework with the self-attention AAM has **88.7%** less trainable parameters when compared to existing approaches. This difference is even greater when scaling to larger models. Training less parameters can reduce the tendency to overfit on the proxy task and

Table 12: Performance when scaling SPLAVU up to larger models.

| Anonymization Method | Model | VISPR cMAP ($\downarrow$) | HMDB51 Top1 Acc. ($\uparrow$) | UCF101 Top1 Acc. ($\uparrow$) | THUMOS14 mAP (%) ($\uparrow$) |
|---|---|---|---|---|---|
| Baseline | InternVideo-H | 74.62 | 79.48 | 98.84 | 62.45 |
| Ours-HMDB51 | | 54.74 | 79.87 | – | 56.35 |
| Ours-UCF101 | | 50.29 | – | 99.21 | 53.28 |
| Baseline | VideoMAEv2-G | 75.69 | 81.05 | 97.81 | 70.09 |
| Ours-HMDB51 | | 53.39 | 80.85 | – | 65.21 |
| Ours-UCF101 | | 51.10 | – | 97.91 | 62.69 |

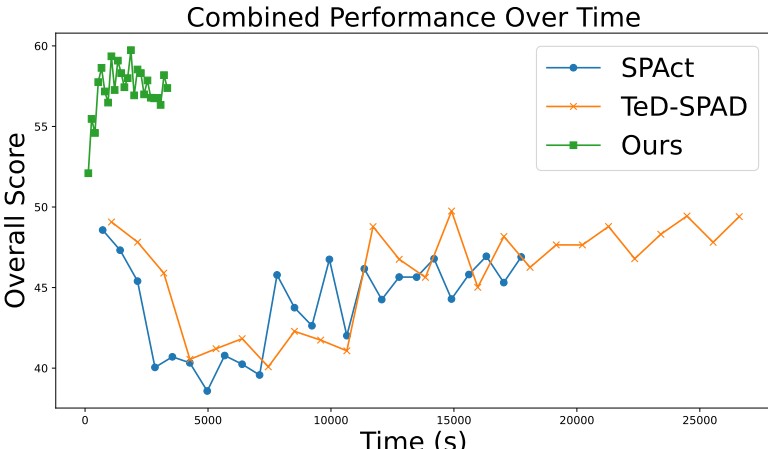

Figure 3: Graph showcasing the overall runtime and accuracy of 3 privacy-preserving methods. The x-axis shows time in seconds and the y-axis has an overall score for accuracy/privacy computed in Equation 16.

allow for learning an effective anonymization on limited training data (see Main Table 4). Also, in federated learning, these parameters are communicated between the server and clients, so the reduced learnable parameters are useful in efficient and privacy-preserving federated learning Zhao et al. (2023); Yu et al. (2022).

The efficiency of our method is further demonstrated using Figure 3. In this instance, our method did not make use of precomputed features, yet it still completed ≈**3.5x** faster than the next fastest method. The combined accuracy/privacy metric is simply defined as follows:

$$y_t = (acc_t + (1 - priv_t)) * 0.5, \tag{16}$$

Table 13: Trainable parameters for each model type/training framework.

| Method | Model | Trainable Parameters (M) |
|---|---|---|
| SPAct/TeD-SPAD SPLAVU | I3D | 55.2 **25.6** |
| SPAct/TeD-SPAD SPLAVU | VideoMAE-B | 129.4 **14.6** |
| SPAct/TeD-SPAD SPLAVU | V-JEPA | 694.2 **39.8** |
| SPAct/TeD-SPAD SPLAVU | InternVideo-H | 675.0 **39.8** |
| SPAct/TeD-SPAD SPLAVU | VideoMAEv2-G | 1055.5 **48.0** |

where $t$ is the current time, $y_t$ is the performance score, and $acc_t$ and $priv_t$ are the top-1 accuracy scores and privacy prediction score using the current $f_A$ model, respectively. Privacy is inverted as a lower private attribute prediction score is considered better. Each method was trained for 50 epochs using the same hyperparameters. The SPLAVU latent anonymization framework achieves a higher, more stable performance at only a fraction of the runtime when compared to input-based methods.

### C.3.2 PRECOMPUTING FEATURE EMBEDDINGS

Since we are using a completely frozen video encoder model $f_E$, the latent feature embeddings can be precomputed for a much faster training process. In this case, only validation augmentations are used, and each video clip is only ran through the model forward pass once. There is flexibility in clip choice and skip rate. In this work, we opt for a simple skip rate of 1 (consecutive frames), and take all non-overlapping sequential clips for each video. The computed embeddings are saved for each video, and a random clip is sampled during training time. The same evenly-spaced 5 video clips are used for validation. Table 14 shows a comparison between using the raw videos and precomputed features. Due to the use of weak augmentations in the raw videos, we see an improvement over using the precomputed. However, using the precomputed features only requires a single forward pass over the dataset, which takes 4 minutes (HMDB51), then only **1.4** minutes for training.

Table 14: Results comparison between AAM trained on HMDB51 using input videos vs. precomputed features. Experiment was done using VideoMAE-B model.

| Training Data | PAP VISPR ($\downarrow$) | AR HMDB51 ($\uparrow$) | TAD T14 ($\uparrow$) | AD UCF-Cr. ($\uparrow$) | Training Time (min) |
|---|---|---|---|---|---|
| Raw Videos | **50.59** | **75.10** | **58.15** | 82.71 | 185.3 |
| Precomp. Features | 54.35 | 73.92 | 56.50 | **84.52** | **4.0+1.4 (5.4)** |

## D  TRAINING ALGORITHM

Algorithm 1 formalizes the SPLAVU workflow notation. We consider anonymizer $f_A$ and task heads $f_{T_{AR}}$, $f_{T_{TAD}}$, and $f_{T_{AD}}$ for the anonymization training and $f_{reco}$, $f_{tad}$, and $f_{wsad}$ for downstream tasks. In order, these models are parameterized by $\theta_A$, $\theta_{T_{AR}}$, $\theta_{T_{TAD}}$, $\theta_{T_{AD}}$, $\theta_{reco}$, $\theta_{tad}$, and $\theta_{wsad}$. $\mathbb{D}_{reco}$, $\mathbb{D}_{tad}$, and $\mathbb{D}_{wsad}$ are all used in the proxy anonymization process, then also for the downstream task evaluation. The downstream $\mathbb{D}_{reco}$ may be the same or different from during the anonymization process.

---

**Algorithm 1:** SPLAVU Framework

---

**1** **Anonymization Training**

**2** **Inputs**:

**3**   *Datasets:* $\mathbb{D}_{reco}$, $\mathbb{D}_{tad}$, $\mathbb{D}_{wsad}$

**4**   *# of Epochs:* $anon\_epochs$

**5**   *Learning Rates:* $\alpha_A$, $\alpha_{AR}$, $\alpha_{TAD}$, $\alpha_{AD}$

**6**   *Hyperparameters:* $\omega_A$, $\omega_T$, $\omega_B$, $\omega_{LC}$, $\omega_{AR}$, $\omega_{TAD}$, $\omega_{AD}$

**7** **Output**: $\theta_A$, $\theta_{T_{AR}}$, $\theta_{T_{TAD}}$, $\theta_{T_{AD}}$

---

**8** Model Initialization:

**9** Initialize $f_E$ with Kinetics400 weights Carreira & Zisserman (2017);

**10** Initialize $\theta_A \leftarrow \theta_A - \alpha_A \nabla_{\theta_A}(\mathcal{L}_{L1}(\theta_A))$

---

**11** Multitask Anonymization Training:

**12** **for** $e_0 \leftarrow 1$ **to** $anon\_epochs$ **do**

**13**  $\quad$ $\theta_A \leftarrow \theta_A - \alpha_A \nabla_{\theta_A}(\omega_{LC}\mathcal{L}_{LC}(\theta_A) + \omega_T \mathcal{L}_{T^*}(\theta_A, \theta_{T_{AR}}, \theta_{T_{TAD}}, \theta_{T_{AD}}) - \omega_B L_B(\theta_A))$

$\quad\quad$ $\theta_{T_{AR}} \leftarrow \theta_{T_{AR}} - \alpha_{AR} \nabla_{\theta_{T_{AR}}}(\mathcal{L}_{AR}(\theta_{T_{AR}}, \theta_A))$,

$\quad\quad$ $\theta_{T_{TAD}} \leftarrow \theta_{T_{TAD}} - \alpha_{TAD} \nabla_{\theta_{T_{TAD}}}(\mathcal{L}_{TAD}(\theta_{T_{TAD}}, \theta_A))$,

$\quad\quad$ $\theta_{T_{AD}} \leftarrow \theta_{T_{AD}} - \alpha_{AD} \nabla_{\theta_{T_{AD}}}(\mathcal{L}_{AD}(\theta_{T_{AD}}, \theta_A))$,

**14** **end**

---

**15** **Downstream Tasks Evaluation**

**16** **Inputs**:

**17**   *Datasets:* $\mathbb{D}_{reco}$, $\mathbb{D}_{anomaly}$, $\mathbb{D}_{tad}$

**18**   *# of Epochs:* $reco\_epochs$, $anomaly\_epochs$, $tad\_epochs$

**19**   *Learning Rates:* $\alpha_{reco}$, $\alpha_{wsad}$, $\alpha_{tad}$

**20** **Output**: $\theta_{reco}$, $\theta_{wsad}$, $\theta_{tad}$

---

**21** Privacy-Preserved Action Recognition Training:

**22** **for** $e_0 \leftarrow 1$ **to** $reco\_epochs$ **do**

**23**  $\quad$ $\theta_{reco} \leftarrow \theta_{reco} - \alpha_{reco} \nabla_{\theta_{reco}}(\mathcal{L}_T(\theta_{reco}, \theta_A))$,

**24** **end**

---

**25** Feature Extraction on $\mathbb{D}_{anomaly}$:

**26** $\mathbb{F}_{anomaly} = \{ f_A(f_E(X^{(i)})) \mid \forall X^{(i)} \in \mathbb{D}_{anomaly} \}$

---

**27** Privacy-Preserved Weakly-Supervised Anomaly Detection (WSAD) Training:

**28** **for** $e_0 \leftarrow 1$ **to** $anomaly\_epochs$ **do**

**29**  $\quad$ $\theta_{wsad} \leftarrow \theta_{wsad} - \alpha_{wsad} \nabla_{\theta_{wsad}}(L_{wsad}(\theta_{wsad}, \mathbb{F}_{anomaly}))$

**30** **end**

---

**31** Feature Extraction on $\mathbb{D}_{tad}$:

**32** $\mathbb{F}_{tad} = \{ f_A(f_E(X^{(i)})) \mid \forall X^{(i)} \in \mathbb{D}_{tad} \}$

---

**33** Privacy-Preserved Temporal Action Detection (TAD) Training:

**34** **for** $e_0 \leftarrow 1$ **to** $tad\_epochs$ **do**

**35**  $\quad$ $\theta_{tad} \leftarrow \theta_{tad} - \alpha_{tad} \nabla_{\theta_{tad}}(L_{tad}(\theta_{tad}, \mathbb{F}_{anomaly}))$

**36** **end**

---

