# OpenReview forum: "Self-supervised Privacy-preservation via Latent Anonymization for Generalizable Video Understanding"
_ICLR.cc/2025/Conference — ICLR 2025 Conference Withdrawn Submission_

### Official Review · Reviewer_CmC5 · 2024-11-02

**Soundness:** 3
**Presentation:** 3
**Contribution:** 2
**Rating:** 5
**Confidence:** 4

**Summary:**

This paper presents an innovative method for privacy preservation in the latent space, designed to be effective across diverse video understanding tasks through a collaborative multitask co-training framework. Unlike traditional approaches that focus on anonymization within the image space, this method shifts the privacy mechanism to the latent space, enhancing privacy while maintaining performance. To achieve this, the paper introduces a self-supervised privacy budget objective based on static clips. Additionally, a latent consistency loss is incorporated to ensure the utility model's generalization capability across tasks.

**Strengths:**

1. The research motivation of the paper is strong, the method is clearly and intuitively designed, and the writing is excellent.
2. The static clip privacy budget objective proposed by the authors is intriguing and presents a novel approach to privacy removal.
3. The experimental results are thorough, and the analysis is comprehensive.
4. The authors have addressed the bias issues related to human attributes, which is commendable.

**Weaknesses:**

1. In previous works, they assumed that video models were managed by public operators or APIs, so they focused on anonymizing the data before inputting it into the video model (encoder). In this paper, however, the authors propose anonymization after the video encoder, which, in my view, means privacy is already exposed when sent to the encoder. This limitation affects the practical applicability of the paper. Could the authors discuss scenarios where latent space anonymization would be preferable or more practical than input-level anonymization, given that the raw data is exposed to the encoder?

2. It appears that this model is trained on multiple tasks and then tested on the same set of tasks, which, in my opinion, is not truly generalized learning but rather multi-task learning. This approach doesn’t seem particularly meaningful since prior methods could also handle multiple tasks—they simply didn’t conduct related experiments. A more valuable approach would be to train on a set of tasks and then generalize to previously unseen tasks.

3. The experimental section misses a comparison with work [1]. Could the authors include a comparison with [1] in their experiments? Overall, I am quite positive about this work; however, these above concerns are important to me. If the authors address them, I would be happy to raise my score.

[1] Joint Attribute and Model Generalization Learning for Privacy-Preserving Action Recognition. NeurIPS 2023.

**Questions:**

Please see weaknesses.

---

### Official Review · Reviewer_yXcF · 2024-11-02

**Soundness:** 1
**Presentation:** 2
**Contribution:** 1
**Rating:** 3
**Confidence:** 3

**Summary:**

This work proposes to tune existing video model to privacy-preserving video model for various down-stream tasks including action recognition, temporal action localization and anomaly detection. While previous works eliminate the privacy information in the pixel level, this work operates in the latent space.

**Strengths:**

1. The proposed method does not require to train a video model from scratch, which saves lots of computational resources.
2. The proposed method supports multiple down-stream tasks while previous methods usually focus on merely one task.

**Weaknesses:**

1. The motivation is unclear and confusing. As depicted in Fig. 1 (a), previous methods suffer from privacy leakage in the utility video model. And in Fig. 1(c), the proposed method also encode the images with visible privacy information using utility video model. Therefore, the proposed method also suffer from the privacy leakage. If the privacy protection is operated in the latent space, the visible privacy information in images cannot be protected, which contradicts the statements in lines 47-49.

2. Limited technical contribution. The proposed Anonymizing Adapter Module is not novel without any insight. The overall architecture is to learn a adapter upon a video model so that the features cannot be directly classified by some attribute classifiers.

3. The comparison results in Table 1 is not sufficient since only two methods are compared.

4. The presentation can be improved. For example, the purpose of the Budget Privacy Objective has not been clearly explained.

**Questions:**

My main concern is the motivation and practicality of this work. Besides, the technical contribution is limited and the experimental results are insufficient.

---

### Official Review · Reviewer_Exsz · 2024-11-04

**Soundness:** 3
**Presentation:** 3
**Contribution:** 2
**Rating:** 3
**Confidence:** 5

**Summary:**

This paper introduces an anonymizing adapter module (AAM) applied in the latent space. The authors utilize minimax optimization to minimize mutual information for privacy while retaining task performance. The method was tested on datasets like Kinetics400, UCF101, HMDB51, THUMOS14, and UCF-Crime, demonstrating computational efficiency and generalization capability on large video foundational models.

**Strengths:**

1. The proposed method shows generalization among the tested tasks (e.g., action recognition, anomaly detection, temporal action detection) without losing significant performance.
2.  SPLAVU is computationally efficient as it does not require full fine-tuning of the utility video encoder.

**Weaknesses:**

1. The definition of privacy is not clear. Also, the privacy evaluation metrics are very unclear. Previous work has extensive discussions on the privacy attributes in videos, which is lacking in this paper.
2. A large part of the related work is under the topic of face de-identification (in videos), which is also not discussed.
3. The generalization relies heavily on the utility loss defined in Eq. (5). However, it has a constraint on the specific tasks. What if the tasks are different, or there are more tasks?

**Questions:**

Please refer the weakness section,

---

### Official Review · Reviewer_fs1Y · 2024-11-04

**Soundness:** 2
**Presentation:** 3
**Contribution:** 2
**Rating:** 5
**Confidence:** 3

**Summary:**

This paper presents a novel method called SPLAVU for achieving privacy protection in video understanding tasks. SPLAVU transfers privacy anonymization from the input features to the latent feature space while maintaining the multitask generalization capability of the video model. The authors' extensive evaluation demonstrates that this method significantly reduces privacy leakage while maintaining near-peak performance across multiple downstream tasks. Additionally, new protocols are proposed to assess and mitigate gender bias in action recognition models.

**Strengths:**

1.This method effectively achieves privacy protection in video understanding tasks while maintaining the multitask generalization capability of the video model.
2.The experimental section is designed comprehensively, covering various scenarios and tasks, and provides ample experimental analysis.

**Weaknesses:**

Limited Novelty: The method lacks uniqueness since it relies on common techniques and uses a framework that primarily combines existing loss functions.

Budget Privacy Objective: There is insufficient explanation about how the Budget Privacy Objective facilitates data anonymization and privacy optimization, and the role of /theta in Equation 6 needs a clearer theoretical analysis.

Inadequate Experimental Analysis: The paper's experimental comparisons with advanced privacy-preserving methods are limited. The analysis could be improved by incorporating more experiments to strengthen the argument, especially in relation to works like STPrivacy and Privacy-Preserving Action Recognition via Motion Difference Quantization.

**Questions:**

1.The paper's novelty is limited since each component of the method is a common technique. The loss function in the framework is primarily a combination of various loss functions.

2.How does the Budget Privacy Objective achieve data anonymization and privacy optimization? In this process, what does \theta represent in Eq. 6? Please provide a more detailed explanation or theoretical analysis.

3. The experimental analysis comparing advanced methods and privacy anonymization is limited, such as references [1] and [2]. I encourage the authors to incorporate more relevant experiments to enhance the paper's persuasiveness. [1] STPrivacy: Spatio-Temporal Privacy-Preserving Action Recognition. [2] Privacy-Preserving Action Recognition via Motion Difference Quantization

4. This raises questions about the model initialization process. Does the video encoder model  only use pretrained model parameters and not undergo any updates in subsequent processes? Additionally, what is the specific process for initializing and updating ? What is the  loss？

---

### Note · Authors · 2024-11-13

**Comment:**

Thank you all for taking the time to review our paper. We really appreciate the positive comments, and we will do our best to improve our work based on the weaknesses. After seeing the initial review scores, we have decided to withdraw and work towards a stronger future submission. Thanks again, best of luck in your efforts!

**Withdrawal Confirmation:**

I have read and agree with the venue's withdrawal policy on behalf of myself and my co-authors.